# Prevalence and associated factors of non-communicable chronic diseases among university academics in Jordan

**Abdullah M. Khamaiseh**[1]*, **Sakhaa S. Habashneh**[2]

**1** Department of Community and Mental Health Nursing, Faculty of Nursing, Mutah University, Mutah, Karak, Jordan, **2** Department of Adult Health Nursing, Faculty of Nursing, Mutah University, Mutah, Karak, Jordan

* akhamaiseh@mutah.edu.jo

**Data Availability Statement:** All relevant data are within the manuscript and its Supporting information files.

**Funding:** The author(s) received no specific funding for this work.

## Abstract

The increasing prevalence of non-communicable chronic diseases on a global, regional, and local scale demonstrates the extensive impact of these diseases, which now account for 70% of all worldwide deaths and affect a diverse population outside affluent places. The purpose of this study was to assess the prevalence and associated factors of non-communicable chronic diseases among academics at Mutah University in Jordan, while also taking into account the global trend of non-communicable diseases impacting different demographics. In a cross-sectional study, the majority of faculty members completed a questionnaire that included demographic information and prevalence of chronic diseases. The most prevalent conditions detected were hypertension (19.6%), diabetes mellitus (17.5%), rheumatoid arthritis (14.2%), heart disease (12.6%), and respiratory disorders (11.3%). Specifically, smoking and being overweight are underlined as significant risk factors, particularly among male respondents. These findings highlight the need of implementing health promotion programs in educational academic institutions, with an emphasis on fostering healthy dietary habits and encouraging physical activity.

## Introduction

The prevalence of non-communicable chronic diseases (NCCDs), which are defined as irreversible pathological alterations affecting particular body systems, has increased globally and is a problem for both developed and developing countries [1].

Almost 70% of deaths worldwide are caused by NCCDs [2–5], with more than 35 million deaths yearly, most of which take place in low- and middle-income countries [6, 7]. These diseases not only lead to functional impairments but also present personal, economic, and social challenges [8]. NCCDs, which account for more than two-thirds of all deaths in the US, are the main causes of mortality, including diabetes, cancer, and cardiovascular diseases [9].

Compelling scientific data indicates a substantial correlation between the occurrence of cancer, cardiovascular disease, chronic respiratory illness, and diabetes and modifiable risk factors. Alcohol intake, obesity, cigarette smoking, unhealthy diet, and physical inactivity all contribute to more than two-thirds of the incidences of these diseases [10].

**Competing interests:** The authors have declared that no competing interests exist.

The prevention and management of NCCDs frequently rely on behavioral interventions, including promoting healthy diet, encouraging higher levels of physical activity, and discouraging unhealthy behaviors such as tobacco use and alcohol use [11].

In Jordan, NCCDs are the primary cause of morbidity and mortality, accounting for 76% of total deaths [12, 13]. Diabetes is prevalent among Jordanians aged 25 and up, accounting for 34% of the population. Furthermore, it was anticipated that the prevalence of chronic diseases in Jordan will quickly increase between 2005 and 2050 [14]. Population-based data on the prevalence of NCCDs is critical for establishing effective preventative and management strategies [11]. Addressing NCCDs and their underlying causes in early adulthood can have a role in delaying disease progression and increasing general well-being throughout an individual's lifespan [15]. NCCDs have a disproportionate impact on the older population worldwide. Statistics show that 80% to 92% of older persons have at least one NCCD, and 50% to 77% have two or more. Cardiovascular disease, chronic obstructive pulmonary disease, and diabetes are the most common non-communicable diseases among the elderly [16].

## Significance of the study

Health and quality of life are greatly impacted worldwide by NCCDs [9]. The increasing prevalence in low and middle-income countries requires immediate attention and a comprehensive response from policymakers and the international public health community [17]. Despite the widespread belief that university academics generally maintain good health [18], Being a researcher training nursing students in hospitals and medical centers exposed a different reality. Academic colleagues regularly receiving monthly medications for various NCCDs were consistently observed. As an associate professor employed in a health faculty, this observation motivates to initiate a study on the health status of academics in similar environments. Consequently, the purpose of this study is to assess the NCCDs and associated factors among academic staff at Mutah University in Jordan. The findings aim to offer valuable insights for intervention planning and stakeholder engagement.

## Literature review

NCCDs, which contribute significantly to morbidity and mortality globally, are a major health concern [19–21]. Lifestyle and dietary changes have made these conditions a major public health issue [22]. Cardiovascular diseases, malignancies, respiratory illnesses, and diabetes are significant NCCDs that are influenced by risk factors such as poor diet, insufficient physical activity, and tobacco use [23]. Preventable risk factors, such as smoking, hypertension, overweight, high cholesterol, alcohol consumption, and improper diet, lead to a significant percentage of NCCDs [20].

In the Eastern Mediterranean Region (EMR), which includes Jordan, NCCDs represent a considerable burden, with mortality rates exceeding those of communicable diseases by more than ten times [24].

## Prevalence of NCCDs and main risk factors in Jordan

Jordan is undergoing an epidemiological shift, with chronic diseases becoming increasingly prevalent. Among Jordanian youths, sedentary lifestyles, poor dietary habits, and smoking are commonly observed health risks. The prevalence of smoking among Jordanian patients with chronic diseases ranges between 21% and 27%, and chronic diseases account for the vast majority of deaths in Jordan. Heart disease and stroke are responsible for more than 30% of these deaths [25]. In comparison to communicable diseases, the mortality rate from non-

communicable diseases in Jordan is nearly 15.5 times higher [25]. These findings highlight the urgent need for preventive measures and comprehensive interventions in the region.

### Research questions

This study aimed to address the following research questions:

What was the prevalence rate of key NCCDs observed within the academics at Mutah University?

What were the factors associated with the primary NCCDs identified among the academic community at Mutah University?

## Materials and methods

### Design and settings

The aim of this study was to assess NCCDs and their associated factors among academic's members at Mutah University in Jordan. To achieve this, a cross-sectional design was utilized to examine the prevalence and determinants of NCCDs among academic staff at Mutah University. The research was carried out at Mutah University encompassing all 15 faculties.

The faculties are categorized into four health-related faculties and eleven non-health related faculties.

### Study participants

The faculty at Mutah University comprises 626 members, contributing to the academic and research endeavors of the institution. The participants in this study comprised the entire academic staff of Mutah University who voluntarily expressed their willingness to participate during the data collection period. The inclusion criteria for the study were as follows:

1. Academic Status: All participants must be currently serving as academic staff at Mutah University.

2. Employment Status: Participants must be employed on a full-time basis.

3. Gender: Both male and female academic staff were eligible for inclusion.

4. Nationality: The study included participants of all nationalities.

5. Consent: Informed consent was a prerequisite for participation in the study.

### Study instrument

The questionnaire employed in the study encompasses two sections:

1. **Demographic Information**: This part gathers fundamental demographic details from participants, including age, gender, weight, height, marital status, education level, academic rank, and faculty affiliation.

2. **Sample Questionnaire for NCCDs**: The subsequent section adopts the Sample Questionnaire for Chronic Disease crafted by the Stanford Patient Education Research Center. This standardized tool gauges the health status of individuals grappling with chronic illnesses across six primary domains:

a). **General Health**: This section typically probes into participants' perceptions of overall health, quality of life, experienced symptoms, and any constraints imposed by health conditions.

b). **Physical Activities**: Queries within this category delve into the frequency, duration, and intensity of participants' physical activities, offering insights into their exercise habits.

c). **Confidence in Task Performance**: This component evaluates participants' confidence levels in executing various tasks pertinent to managing their health condition, such as adhering to treatment plans, coping with symptoms, and communicating effectively with healthcare providers.

d). **Daily Activities**: Questions here explore how NCCDs impacts participants' daily routines, encompassing work, household chores, social engagements, and recreational pursuits.

e). **Medical Care**: This segment assesses participants' satisfaction with the medical care they receive, their access to healthcare services, the quality of communication with healthcare providers, and their overall healthcare experiences.

The Sample Questionnaire for NCCDs from the Stanford Patient Education Research Center typically comprises multiple-choice queries, Likert scale items, and open-ended questions. Response options span a spectrum from "strongly agree" to "strongly disagree," aiming to capture a broad range of experiences related to living with NCCDs, including physical well-being, emotional health, functional abilities, and interactions with healthcare services. Utilizing this standardized tool allows for comparability with other studies using similar instruments, contributing to a broader understanding of NCCDs across diverse populations. All the original questionnaire items from Stanford Patient Education Research Center used in the data collection. However, as the focus of this manuscript was solely on questions that were directly relevant to the research title, objectives, and questions, certain items were overlooked by data analysis.

## Instrument validity

The questionnaire was subjected to a validity assurance method that includes translation into Arabic by an experienced professional translator with extensive experience translating healthcare and medical research content. This translator displayed a high level of expertise in understanding the terminology and concepts pertaining to healthcare and related sectors contained within the questionnaires, followed by back translation into English. A comprehensive comparison of the post-translated English version and the original confirmed that they were identical. In addition, a pilot study was carried out with 30 faculty members. The goal was to determine the feasibility of the study protocol, ensure the questionnaire's comprehensiveness, confirm the clarity and consistency of the questions, and make any required changes to the questionnaire's language. It's important to highlight that the findings of the pilot study were not incorporated into the present study.

## Procedure

The Stanford Patient Education Research Center provided permission to utilize the complete questionnaire or any of its portions without fee. Before beginning data collection and after receiving official consent from the university administration, extensive briefings were held with deans and department heads from each selected faculty to cover all aspects of the study. They received questionnaires, information sheets, letters of ethical approval, and permission

forms. Six research assistants were recruited and trained prior to the start of the study to help with the research. The responsibilities of the research assistants included distributing questionnaires, addressing queries, and providing support to participants who needed assistance in completing the questionnaires. Subsequently, the research assistants collected the completed questionnaires and handed them over to the researchers. The data collection phase extended for a month, starting on Sunday, February 5, 2023, and concluding on Sunday, March 5 of the same year.

### Ethical approval

The study adhered to the principles expressed in the Declaration of Helsinki and received approval from the Faculty of Nursing's Institutional Review Board under Proposal No. EC2/2018, in accordance with Mutah University's ethical norms. Participants were given an Arabic-language consent form that outlined the study's aims, guaranteed anonymous data collection, and ensured the highest level of confidentiality. Participation in the study was voluntary, and staff members were encouraged to signify their consent through both formal written and informal verbal channels.

### Statistical analysis

Data analysis was performed using IBM SPSS Statistics for Windows, version 25.0, developed by IBM Corp. Descriptive statistics, such as frequencies, percentages, means, and standard deviation, were employed to present the demographic characteristics of the participants. Furthermore, inferential statistics were applied, utilizing the chi-square test, with significance set at the 0.05 level.

## Results

A total of 412 questionnaires were distributed to participants who met the eligibility criteria in the various faculties, that two hundred and forty-one (241) questionnaires were completed and returned to the researcher, yielding a response rate of 58%.

The results comprised demographic characteristics, the prevalence of NCCDs, and factors associated with these NCCDs.

### Demographic characteristics of the study participants

Table 1 outlines the demographic information of the 241 academics who took part in the study. A significant majority of the participants were male (n = 189; 78.4%). The mean age of the study participants was 44.50±9.60 years, covering an age range from 25 to 70 years. Almost half of the participants were in the age bracket of 41 to 60 years, and around one-third held the role of assistant professor. The prevailing marital status among the subjects was married, accounting for 89.6% of the participants.

As shown in Fig 1. Hypertension (19.6%), diabetes mellitus (DM) (17.5%), rheumatic arthritis (14.2%), cardiac diseases (12.6%), and respiratory disorders (11.3%) were the most common Non-Communicable Diseases(NCCDs) among academics. Male academics exhibited a higher prevalence of these five NCCDs compared to their female counterparts.

Fig 2 illustrates that the rating scale ranged from "None"(0) to Severe pain (7–10). Approximately 62.5% of the academics noted their typical pain interfering with their activities as mild, followed by 27.6% experiencing moderate pain, 5.5% experiencing severe pain, and 4.3% reporting no pain.

**Table 1. Participants demographic characteristics.**

| Variable | Gender | | Total | P- Value |
|---|---|---|---|---|
| | Male | Female | | |
| **Age** | N (%) | N (%) | | 0.000 |
| 20–30 | 5(2.1) | 8(3.3) | 13(5.4) | |
| 31–40 | 55(23) | 27(11.3) | 82(34.3) | |
| 41–60 | 104(43.5) | 15(6.3) | 119(49.8) | |
| >60 | 23(9.6) | 2(0.8) | 25(10.4) | |
| **Rank** | | | | |
| Full prof | 56(23.2) | 4(1.7) | 60(24.9) | 0.000 |
| Associate prof | 47(19.7) | 5(2.1) | 52(21.8) | |
| Assistant Prof | 60(25.1) | 19(7.9) | 80(33.0) | |
| Teacher | 14(5.9) | 8(3.3) | 22(9.2) | |
| Clinical Instructor | 3(1.2) | 3(1.2) | 6(2.5) | |
| Bachelor | 7(2.9) | 14(5.8) | 21(8.7) | |
| **Marital status** | | | | |
| Married | 178(74.2) | 37(15.8) | 215(89.6) | 0.000 |
| Single | 8(3.3) | 14(5.8) | 22(9.2) | |
| Divorced | 1(0.4) | 1(0.4) | 2(0.8) | |
| Widowed | 1(0.4) | 0(0.0) | 1(0.4) | |
| **Faculty** | | | | |
| Health faculties | 86(37.2) | 15(6.5) | 101(43.7) | 0.037 |
| Non health faculties | 96(41.6) | 34(14.7) | 130(56.3) | |

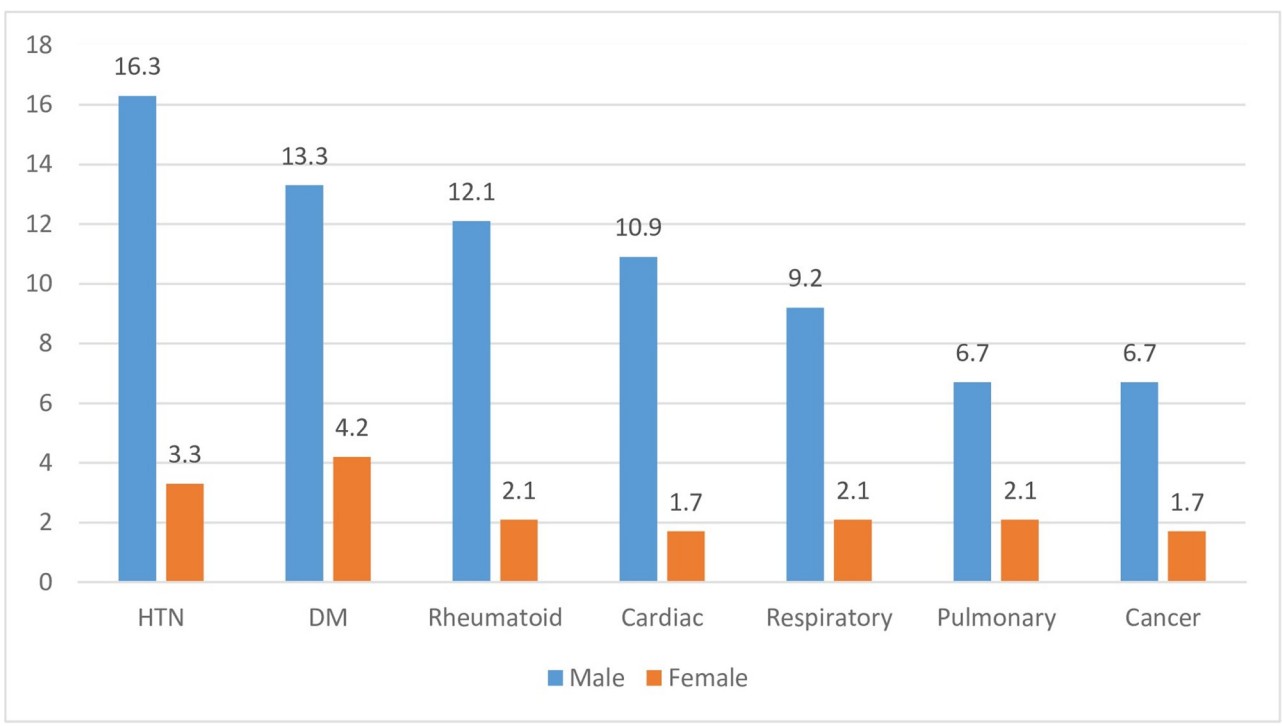

**Fig 1. Prevalence of chronic diseases among the participants.** Hypertension(HTN) (19.6%), diabetes mellitus (DM) (17.5%), rheumatic arthritis (14.2%), cardiac diseases (12.6%), and respiratory disorders (11.3%) were the most common Non-Communicable Chronic Diseases (NCCDs) among academics. Male academics exhibited a higher prevalence of these five NCCDs compared to their female counterparts.

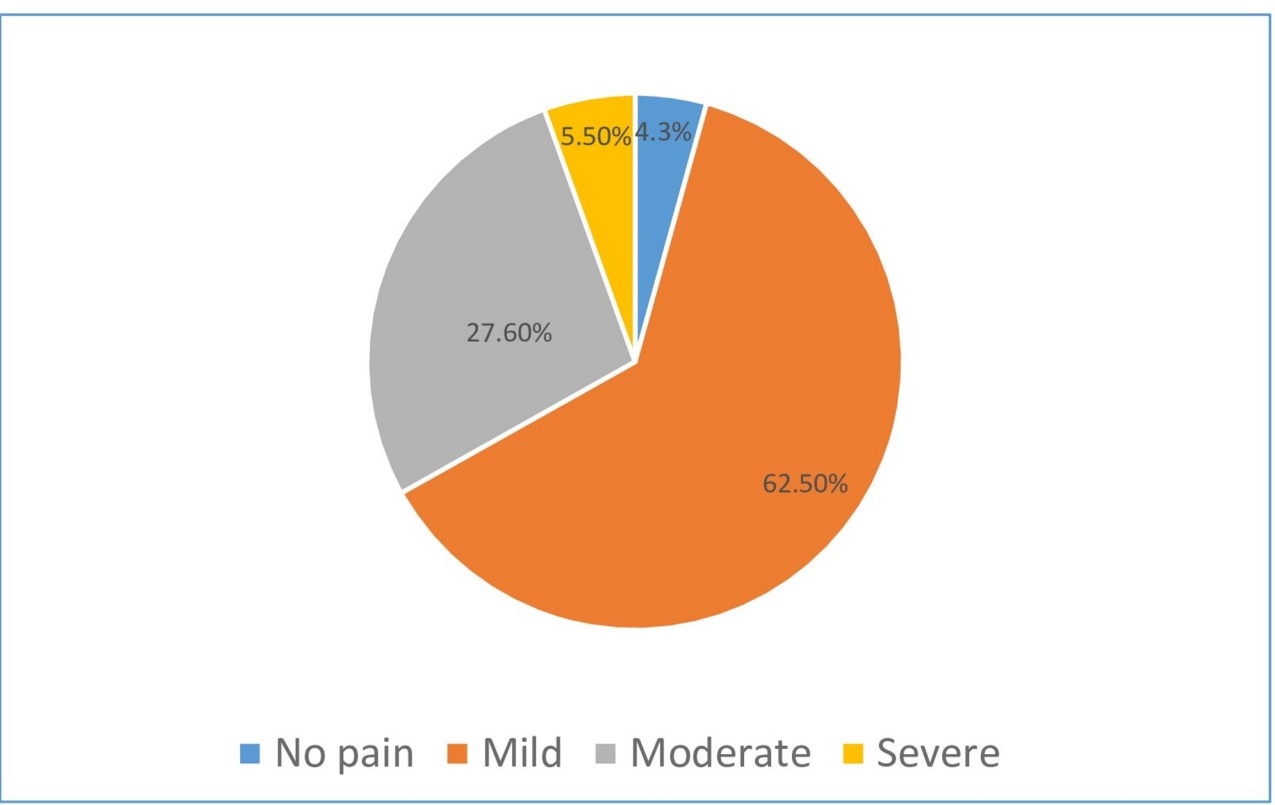

**Fig 2. The Numeric Pain Rating Scale (NPRS).** The rating scale ranged from "None" (0) to "Severe pain" (7–10). Approximately 62.5% of the academics noted their typical pain interfering with their activities as mild, followed by 27.6% experiencing moderate pain, 5.5% experiencing severe pain, and 4.3% reporting no pain.

According to Fig 3, approximately half of the participants (53.1%) viewed their health status as very good, 22.2% as excellent, and 21.8% as good, with only 2.9% describing their health as fair.

### Prevalence of risk factors among participants according to gender

Regarding the associated risk factors, the data in Table 2 showed that 12.3% of men and 2.6% of women were obese, while 44.9% of men and 9.3% of women were overweight. Smoking prevalence is much higher among men than women (32.7,1.8 respectively). Table 2 also displays that 15% of men and 3.4% were never performing walking exercises during the last week of the study. It is noteworthy that individuals who answered the question expressed that walking more than three hours a week, is considered regular walking Finally, just as men were more than women in the prevalence of NCCDs, the same situation with regard to risk factors.

### Discussion

The purpose of this cross-sectional study was to assess the prevalence of NCCDs and associated factors among the university's academic staff. There is no literature on academics to which the findings of this study can be compared. As a result, the comparison was made with different populations described in previous studies.

Among the 241 participants, the study revealed that the most prevalent NCCDs among academics were hypertension (19.6%), diabetes mellitus (17.5%), rheumatoid arthritis (14.2%),

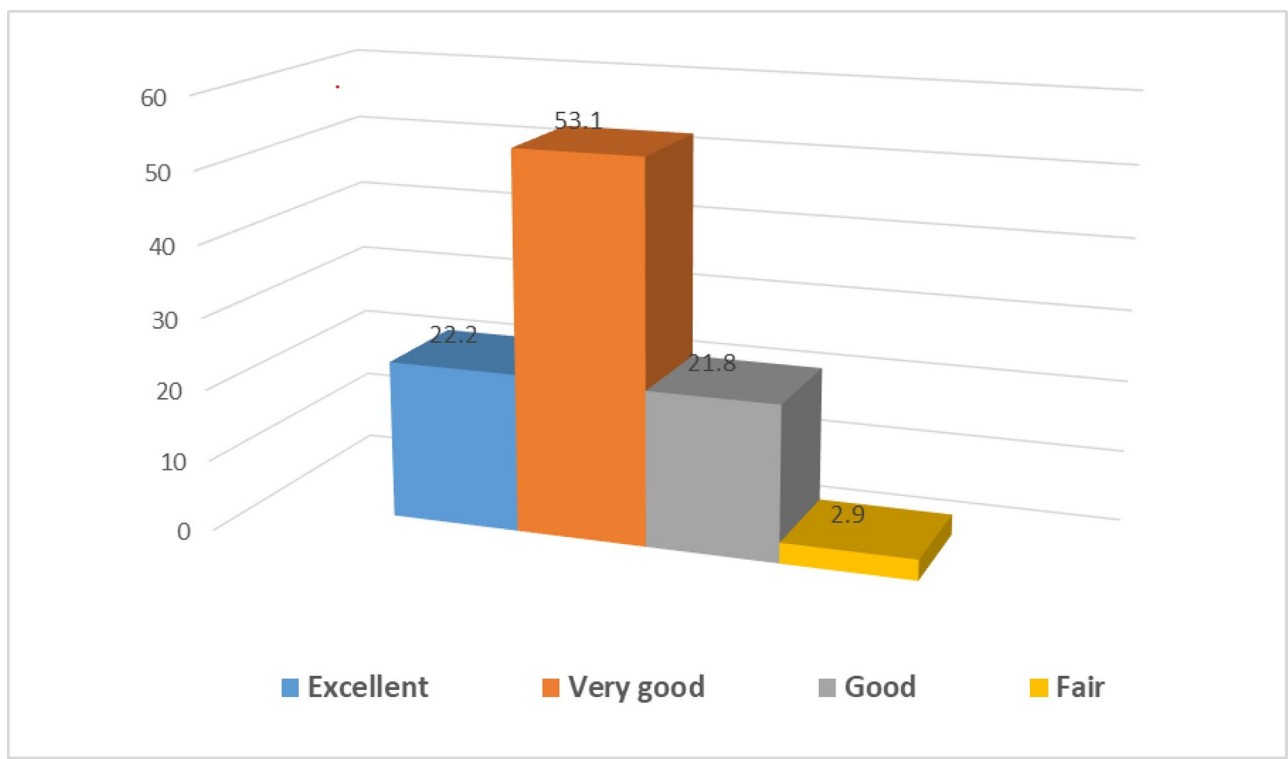

**Fig 3. Academics health perception.** Approximately half of the participants (53.1%) viewed their health status as very good, 22.2% as excellent, and 21.8% as good, with only 2.9% describing their health as fair.

cardiac diseases (12.6%), and respiratory disorders (11.3%). These findings are consistent with a study conducted in Addis Ababa, where chronic arthritis, hypertension, and diabetes mellitus were reported as the most common diseases among participants [6]. In contrast, the prevalence in our study is lower than a study in Saudi Arabia, where 71.3% had hypertension, 27.3% had diabetes, 16.4% had heart disease, and 9.7% had asthma. This difference may be attributed to cultural, custom, and lifestyle disparities between the two countries. Additionally, the variation could be influenced by the different levels of education, as our study focused on academics, while the Saudi study included adults in general. A study in Amman, Jordan, aimed at identifying individuals at elevated risk of developing type 2 diabetes, reported that 37% of participants had hypertension, and 28% had diabetes mellitus [26]. The higher prevalence in the Jordanian study might be associated with a genetic factor, considering that more than two-thirds of the participants had a direct relative diagnosed with diabetes. Our study indicated lower rates of hypertension and diabetes compared to a study in Palestine, where

**Table 2. Prevalence of risk factors among participants according to gender.**

| Risk factor | Male | Female | Total | P-value |
|---|---|---|---|---|
| Obesity | 28(12.3) | 6(2.6) | 34(14.9) | .009 |
| Overweight | 102(44.9) | 21(9.3) | 123(54.2) | .009 |
| Never walking regularly | 35(15) | 8(3.4) | 43(18.4) | .0966 |
| Smoking | 74(32.7) | 4(1.8) | 78(34.5) | 0.000 |

cardiovascular disease prevalence was 28.4% and 19.1%, respectively [27]. The prevalence of heart disease was also lower in our study (8.3% compared to 12.6%).

Furthermore, the prevalence of hypertension in our study is lower than in studies conducted in Swaziland and Saudi Arabia, where rates were 48.3% and 39.2%, respectively [3, 28]. Alshloul suggested that individuals with higher education levels, such as doctorate and master holders, may have lower vulnerability to hypertension due to increased awareness of healthy lifestyle practices [28].

A recent study in Jordan aimed to evaluate the prevalence of obesity among Jordanian women and its relationship with several NCCDs. The findings revealed that 59.8% of participants reported arthritis-related pain. However, the most frequent NCCDs observed were hypertension (29.5%), followed by diabetes and hypertriglyceridemia to a lesser extent [14].

Addressing obesity as a significant risk factor, our study revealed a general prevalence of obesity and overweight at 14.9% and 54.2%, respectively. These findings differ from a study in the UAE, where the prevalence of overweight and obesity was 43.0% and 32.3%, respectively [29]. Differences in study populations and education levels could contribute to these variations. The prevalence of overweight and obesity is increasing globally, both in developed and developing countries. Obesity rates were recorded at 46% in Kuwait, 35% in the UAE, and 34% in Bahrain [30]. Maintaining a healthy body weight is critical for avoiding the health concerns associated with overweight and obesity, thereby lowering both morbidity and mortality [30]. Our study found that males were more likely than females to be overweight or obese.

Regarding smoking prevalence, our study showed that 32.7% of males and 1.8% of females reported smoking. This contrasts with a study in Jordan, where the overall prevalence of cigarette smoking was 59.1% among males and 23.3% among females [31]. In Jordanian society, smoking is more commonly accepted among men than women, often due to societal ideals of masculinity and femininity. Smoking might signify status or defiance for men, whereas women often encounter greater social disapproval. Furthermore, variances in health behaviors between genders are influenced by upbringing, healthcare accessibility, and individual perceptions of health risks. Men and women may have different coping strategies for stress and levels of readiness to get help quitting smoking. Furthermore, creating health initiatives that are individually aimed at each gender may have different results in terms of smoking behaviors. Cultural norms that discourage women from smoking may have contributed to the decreased frequency among Jordanian women in our study [32], and female participants might provide false information in their responses.

Our study revealed that 81.6% of academic staff members walk regularly—that is, they walk for more than three hours a week—despite their busy schedules, which include teaching, research, administrative work, and other responsibilities.

This rate exceeds the 12.5% reported in a national study in Jordan [33], reflecting a higher level of physical activity among academics. This increased activity could be due to their understanding of its role in preventing NCCDs. Furthermore, the availability of suitable facilities and walking paths on campus may contribute to this elevated level of physical activity among academics. Regular physical activity is recognized as a healthy behavior that reduces the risk of NCCDs [33].

## Limitations

This research takes the form of an observational cross-sectional study, which poses challenges in establishing causal relationships with NCCDs. Moreover, Recognizing the constraints of subjective questionnaire items and the biases they might bring forth allows researchers to actively refine research methodologies. This understanding can motivate future investigations

to seek alternative strategies, like mixed methods research or triangulating data sources, in order to mitigate subjectivity's impact and improve the validity of research outcomes. Moreover, there is no literature specifically on academics to compare the results of this study with. Therefore, the comparison applied with various populations reported in other studies means that the findings of this study do not necessarily representative to all academics in Jordan's public universities.

## Implications

Studying the prevalence and associated factors of NCCDs among university academics in Jordan holds significant implications for health professions across education, practice, and research.

In education, understanding the prevalence and contributors to NCCDs can raise awareness among students and faculty about preventive health measures. This knowledge can be integrated into curricula, focusing on lifestyle medicine, preventive care, and occupational health, tailored to university settings. Additionally, promoting healthy habits and offering regular health screenings can be part of educational initiatives.

In practice, universities can establish occupational health programs to address academics' specific health needs, including health screenings, ergonomic assessments, and mental health support. Health promotion initiatives, such as wellness programs and access to fitness facilities, can be implemented, fostering multidisciplinary collaboration among healthcare professionals, educators, and administrators.

In research, investigating the risk factors contributing to NCCDs among academics can inform tailored interventions. Intervention studies can evaluate the effectiveness of strategies like lifestyle interventions and workplace wellness programs. Longitudinal studies tracking health outcomes can provide insights into the trajectory of NCCDs and the efficacy of interventions.

In summary, understanding NCCDs among university academics in Jordan can guide educational efforts, practice guidelines, and research agendas. By integrating this knowledge, implementing targeted interventions, and conducting research, health professions can contribute to promoting the health and well-being of academics and reducing the burden of NCCDs in this population.

## Conclusions

The most prevalent NCCDs among academics were hypertension, Diabetes Mellitus, Rheumatoid arthritis, Cardiac diseases, and respiratory disorders. Men exhibited higher rates of overweight and smoking compared to women, posing significant risk factors. Implementing wellness initiatives within universities that encourage healthy lifestyle choices, including proper dietary habits and regular physical activity, is strongly recommended. Additionally, offering health promotion programs that address modifiable risk factors such as smoking, screening for common NCCDs, and providing treatment and follow-up for affected individuals are essential to mitigate the impact of these diseases. Nevertheless, it is crucial to conduct future analytical epidemiological studies to thoroughly investigate the prevalence and risk factors of NCCDs among academics. Specific suggestions for future research and interventions aimed at promoting health and reducing NCCDs rates among academic staff could include conducting longitudinal studies to evaluate the impact of workplace wellness programs tailored to this population. Additionally, implementing educational workshops focusing on nutrition and stress management, along with incorporating regular physical activity breaks into the daily work schedule, presents promising strategies for exploration and

implementation. These targeted initiatives not only have the potential to enhance the health and well-being of academic staff but also to cultivate a supportive and health-conscious work environment.

## Supporting information

**S1 File. Stanford patient education questionnaire.**
(PDF)

**S2 File. Arabic study questionnaire.**
(DOC)

**S3 File. English Study questionnaire.**
(DOC)

## Acknowledgments

The researchers would like to express their heartfelt gratitude to the university administration, Faculties deans, and department heads for their assistance in gathering data, as well as to all the academics who took part in the study.

## Author Contributions

**Conceptualization:** Abdullah M. Khamaiseh, Sakhaa S. Habashneh.

**Data curation:** Abdullah M. Khamaiseh.

**Formal analysis:** Abdullah M. Khamaiseh.

**Investigation:** Abdullah M. Khamaiseh.

**Methodology:** Abdullah M. Khamaiseh.

**Project administration:** Sakhaa S. Habashneh.

**Resources:** Sakhaa S. Habashneh.

**Writing – original draft:** Abdullah M. Khamaiseh.

**Writing – review & editing:** Abdullah M. Khamaiseh, Sakhaa S. Habashneh.

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
