## [Decision Letter · Decision Letter 0]

18 Mar 2024

PONE-D-24-06204Prevalence of and factors associated with chronic diseases among university academics in JordanPLOS ONE

Dear Dr. Khamaiseh,

Thank you for submitting your manuscript to PLOS ONE. After careful consideration, we feel that it has merit but does not fully meet PLOS ONE’s publication criteria as it currently stands. Therefore, we invite you to submit a revised version of the manuscript that addresses the points raised during the review process.

We look forward to receiving your revised manuscript.

Kind regards,

Ala'a B. Al-Tammemi, MD, MPH

Academic Editor

PLOS ONE

Journal Requirements:

Reviewers' comments:

Reviewer's Responses to Questions

**Comments to the Author**

1. Is the manuscript technically sound, and do the data support the conclusions?

Reviewer #1: Yes

Reviewer #2: Yes

Reviewer #3: Yes

2. Has the statistical analysis been performed appropriately and rigorously? 

Reviewer #1: Yes

Reviewer #2: No

Reviewer #3: Yes

3. Have the authors made all data underlying the findings in their manuscript fully available?

Reviewer #1: Yes

Reviewer #2: Yes

Reviewer #3: Yes

4. Is the manuscript presented in an intelligible fashion and written in standard English?

Reviewer #1: Yes

Reviewer #2: Yes

Reviewer #3: Yes

5. Review Comments to the Author

**Reviewer #1: Suggestions**

1. I suggest to edit the title as follow: Prevalence and Associated Factors of Chronic Diseases Among University Academics in Jordan.

2. Line 146: The term "subjects" may not be appropriate in this context, as it can imply a level of passivity. Consider using "Participants" or "Study Participants" for clarity and to maintain professionalism.

3. Line 162: Provide more detail about the self-administered structured questionnaire, such as the number of questions, response options, and any specific chronic diseases addressed. This will enhance the understanding of the instrument's content and relevance to the study.

4. Line 174: Specify the qualifications and expertise of the professional translator who translated the questionnaire into Arabic to ensure accuracy and reliability in the translation process.

5. Line 202: Specify whether the written consent form provided to participants was available in both Arabic and English to accommodate diverse language preferences among the academic staff.

COMMENT 1: Although the study mentions the participation of a majority of faculty members, it is crucial to include specifics about the sample size and the proportion of faculty members who were approached but declined to participate. Furthermore, the study should address whether the sample accurately represents the broader academic population at Mutah University.

COMMENT 2: The specified timeline for data collection (February 5 to March 5, 2023) provides clarity regarding the study's duration. However, it would be beneficial to discuss any considerations regarding seasonal variations or academic schedules that could impact participant availability or response rates during this period.

COMMENT 3: The discussion underscores significant gender disparities in smoking prevalence, noting a higher percentage of male participants reporting smoking compared to females. However, it would be beneficial to explore the underlying reasons for these disparities further, taking into account factors such as cultural norms, social attitudes towards smoking, and gender-specific health behaviors.

**Reviewer #2: Thank you for the opportunity to review your good work. However, every review brings comments to improve the work. Please consider the following comments:**

Lines 50-53: The two sentences in lines 50-53 might be redundant. Please review them.

Lines 64-67: This part includes a long confusing sentence that needs to be rewritten.

Lines 76-77: A reference is needed for the following sentence “Despite the widespread belief that university academics generally maintain good health”.

Lines 96-98: the sentence is lengthy, unclear, and includes a new abbreviation NCDs!

Lines 106-110: Thesis-like summary at the end of the introduction is undesirable.

Lines 111-117: Subtitles under the research questions are not needed. Be more concise and just mention the research questions.

Lines 118-122: Once again, a thesis-like summary is not appropriate in research articles.

Methods

Line 131: Do we need to know the area of the university, does it matter?

Lines 132-137: Instead of mentioning the faculties by name, it is worth noting that classifying faculties into health and non-health faculties might bring up some significant and more meaningful differences.

Lines 138-145: This paragraph introduces extra information about students, better to remove.

General comments:

Although the researchers studied academics in Jordan who are apparently older than 40 years old, the introduction introduces the prevalence of chronic diseases among young people. The introduction needs to be enriched with more information and statistics about NCCDs among older people.

Throughout the manuscript, the researchers keep mentioning chronic diseases despite introducing the NCCDs term at the beginning of the introduction. Can you use the term NCCDs instead of just repeating the names of chronic diseases?

Methods:

Did you use a priori or post hoc analysis to estimate the sample size? Doing so is necessary to draw valid conclusions.

Results:

Line 240: Suddenly, a new term (Pain) was introduced! This is very far from the study's aims.

More data analysis is needed, the current results display just descriptive statistics.

Table 1: it was mentioned in the methodology that both Jordanian and Non-Jordanian academics were included in the study. However, table 1 doesn't show this classification.

Subheadings under the results are preceded by unnecessary words (part1, part2, and part 3).

Did you follow the STROBE checklist in reporting your research? Please add it as a supplementary file.

The quality of figure one is poor, a new figure with high resolution is needed to be produced. Reference list: Mistakenly, a full stop appears before the number of each reference.

**Reviewer #3: Dear Authors,**

Thank you for your efforts in conducting a very interesting manuscript. Please take all of these comments and recommendations into consideration in your decision to enhance the paper, as follows:

1- The discussion identifies a high prevalence of regular walking among academic staff, attributing it to awareness of the importance of physical activity in preventing chronic diseases. While this interpretation is plausible, further discussion on potential barriers to physical activity among academic staff, such as time constraints or work-related stress, would provide a more nuanced understanding of the findings.

2- The discussion briefly mentions the importance of maintaining a healthy body weight and engaging in regular physical activity to prevent chronic diseases. However, offering specific recommendations for future research directions and interventions aimed at promoting health and reducing the prevalence of chronic diseases among academic staff would enhance the discussion's practical implications.

3- It is noted that specific items in the questionnaire are subjective and rely on each participant's individual opinions. While subjectivity in questionnaire items is common, it's essential for researchers to acknowledge this limitation and consider potential biases introduced by subjective responses.

4- The implications could explore how the study findings could be integrated into the nursing education curriculum to prepare future nurses for addressing chronic diseases in community settings.

6. PLOS authors have the option to publish the peer review history of their article (what does this mean?). If published, this will include your full peer review and any attached files.

Reviewer #1: No

Reviewer #2: No

Reviewer #3: No

---

## [Author Response · Author response to Decision Letter 0]

8 Apr 2024

I have thoroughly examined all the feedback provided by the reviewers, and I am confident that my response adequately addresses their comments.

---

## [Decision Letter · Decision Letter 1]

23 Apr 2024

PONE-D-24-06204R1Prevalence and Associated Factors of Non-communicable chronic diseases Among University Academics in Jordan.PLOS ONE

Dear Dr. Khamaiseh,

Thank you for submitting your manuscript to PLOS ONE. After careful consideration, we feel that it has merit but does not fully meet PLOS ONE’s publication criteria as it currently stands. Therefore, we invite you to submit a revised version of the manuscript that addresses the points raised during the review process.

We look forward to receiving your revised manuscript.

Kind regards,

Ala'a B. Al-Tammemi, MD, MPH

Academic Editor

PLOS ONE

**Additional Editor Comments:**

1- Many paragraphs of the manuscript seem like AI-Assisted Language/writing, please confirm if AI software was used or not.

2- The manuscript needs extensive English language editing.

3- Line 95-100: Rephrase the sentence without using first person pronouns

4- Lines 104 – 126: I do not understand the rationale behind these sections as they both contains repetitive information which was already provided in the introduction

5- Study instruments: The authors explain that they adopted the Sample Questionnaire for Chronic Disease from Stanford Patient Education Research Center. Did the author use part or all the questionnaire items? The authors should clarify this fact and they have to explain to the readers the operationalization of the only items used or adopted from the questionnaire to avoid confusion.

How did the authors measure BMI to identify obesity vs overweight? Did the survey include body weight and height? Also, why alcohol use was not examined as it is a potential risk factor? Also, how did the author define regular walking? Any criteria?

6- Discussion Section: While comparing the findings with regional studies, are the comparison between countries involves studies among academics? Comparing your study findings which is a single-center study among academics with other regional studies that involve various study populations will not be reliable. Please modify the discussion to compare your findings with regional or international studies that examined NCCDs among academic staff.

7- Limitations: Line 423-424: Better to be omitted

8- Implications: Unfortunately, the implications provided do not match the overall objective of the study. Why would the authors direct the implications toward nurses’ education only? I know that this was respectfully suggested by one of the reviewers, however, discuss in this section the potential consequences, recommendations, or practical applications of research findings without specifying nurses’ roles per se.

9- The authors are attaching the Stanford Survey only. Please provide the survey that the authors formulated to collect the data in this study for replicability and not the Stanford questionnaire (as it is already available publicly).

Reviewers' comments:

Reviewer's Responses to Questions

**Comments to the Author**

1. If the authors have adequately addressed your comments raised in a previous round of review and you feel that this manuscript is now acceptable for publication, you may indicate that here to bypass the “Comments to the Author” section, enter your conflict of interest statement in the “Confidential to Editor” section, and submit your "Accept" recommendation.

Reviewer #1: All comments have been addressed

Reviewer #2: (No Response)

Reviewer #3: All comments have been addressed

2. Is the manuscript technically sound, and do the data support the conclusions?

Reviewer #1: Yes

Reviewer #2: Yes

Reviewer #3: Yes

3. Has the statistical analysis been performed appropriately and rigorously? 

Reviewer #1: Yes

Reviewer #2: Yes

Reviewer #3: Yes

4. Have the authors made all data underlying the findings in their manuscript fully available?

Reviewer #1: Yes

Reviewer #2: Yes

Reviewer #3: Yes

5. Is the manuscript presented in an intelligible fashion and written in standard English?

Reviewer #1: Yes

Reviewer #2: Yes

Reviewer #3: Yes

6. Review Comments to the Author

Reviewer #1: Dear Dr,

I have carefully revised the commentary provided in the response file and thoroughly reviewed the revised manuscript. I am pleased to inform you that all the comments and suggestions provided have been diligently addressed and incorporated into the updated version. Therefore, I believe that no further revisions or comments are necessary at this time.

Thank you for your time and guidance throughout this process. I am confident that the manuscript is now ready for publication.

Best regards,

Reviewer #2: Thank you for addressing most of the concerns and comments I raised. However, the article still lacks for the STROBE checklist with proper citation.

Reviewer #3: I appreciate all of your hard work. It appears that every suggestion and comment was taken into consideration.

7. PLOS authors have the option to publish the peer review history of their article (what does this mean?). If published, this will include your full peer review and any attached files.

Reviewer #1: **Yes: **Nour Amin Elsahoryi

Reviewer #2: No

Reviewer #3: No

---

## [Author Response · Author response to Decision Letter 1]

4 May 2024

I express my sincere gratitude to the editor for his insightful comments. I'm hoping my responses to them will be acceptable and persuasive.

---

## [Editor Report · Decision Letter 2]

20 May 2024

Prevalence and Associated Factors of Non-communicable chronic diseases Among University Academics in Jordan.

PONE-D-24-06204R2

Dear Dr. Abdullah Mousa Khamaiseh

We’re pleased to inform you that your manuscript has been judged scientifically suitable for publication and will be formally accepted for publication once it meets all outstanding technical requirements.

Kind regards,

Ala'a B. Al-Tammemi, MD, MPH

Academic Editor

PLOS ONE

Additional Editor Comments (optional):

Dear Dr.Abdullah Mousa Khamaiseh,

Thank you for this hard work and for addressing all the reviewers' and academic editor's comments.

Best regards
---

## [Editor Report · Acceptance letter]

3 Jun 2024

PONE-D-24-06204R2 

PLOS ONE

Dear Dr. Khamaiseh, 

I'm pleased to inform you that your manuscript has been deemed suitable for publication in PLOS ONE. Congratulations! Your manuscript is now being handed over to our production team.

Kind regards, 

on behalf of

Dr. Ala'a B. Al-Tammemi 

Academic Editor

PLOS ONE